# Doubly Stochastic Adversarial Autoencoder

## Abstract

Any autoencoder network can be turned into a generative model by imposing an arbitrary prior distribution on its hidden code vector. Variational Autoencoder uses a KL divergence penalty to impose the prior, whereas Adversarial Autoencoder uses generative adversarial networks. A straightforward modification of Adversarial Autoencoder is to replace the adversary by maximum mean discrepancy (MMD) test. This replacement leads to a new type of probabilistic autoencoder, which is also discussed in our paper.

However, an essential challenge remains in both of these probabilistic autoencoders, namely that the only source of randomness at the output of encoder, is the training data itself. Lack of enough stochasticity can make the optimization problem non-trivial. As a result, they can lead to degenerate solutions where the generator collapses into sampling only a few modes.

Our proposal is to replace the adversary of adversarial autoencoder by a space of *stochastic* functions. This replacement introduces a a new source of randomness, which can be considered as a continuous control for encouraging *explorations*. This prevents the adversary from fitting too closely to the generator and therefore leads to more diverse set of generated samples. Consequently, the decoder serves as a better generative network, which unlike MMD nets scales linearly with the amount of data. We provide mathematical and empirical evidence on how this replacement outperforms the pre-existing architectures.

## 1 Introduction

Any autoencoder network can be turned into a generative model by imposing an arbitrary prior distribution on its hidden code vector. Variational Autoencoder (VAE) [2] uses a KL divergence penalty to impose the prior on the hidden code vector of the autoencoder, whereas Adversarial Autoencoder (AAE) [1] uses generative adversarial networks (GAN) [2]. GAN trades the complexities of *sampling* algorithms with the complexities of *searching* Nash equilibrium in minimax games. Such minimax architectures get updated directly with the help of data examples and gradients flowing through a generative and an adversary. A straightforward modification of AAE is to replace the adversarial network by maximum mean discrepancy (MMD) net [4-5]. This replacement leads to a new type of probabilistic autoencoder, which is also discussed in our paper.

However, an essential challenge remains in both of these probabilistic autoencoders, namely that the only source of randomness at the output of encoder, is the training data itself. Lack of enough stochasticity can make the minimax search non-trivial. As a result, it can lead to degenerate solutions where the generator collapses into sampling only a few modes.

In order to mitigate the mode collapse issue, we introduce *randomness* to the loss functions of such minimax architectures, which can be considered as a continuous control for encouraging *explorations*. This prevents the adversary from fitting too closely to the generator and therefore leads to more diverse set of generated samples. Introducing randomness at the high dimensional feature space can make the minimax search problem very challenging. Fortunately, this is a much easier search problem in our case, thanks to the lower dimensionality of latent space (encoder output).

## 2 PROBABILISTIC AUTOENCODERS

We consistently refer to any random variable and their realization with calligraphic fonts and small letters respectively.

Fix an autoencoder (AE) with encoder parameters of $\theta \in \Theta$ that gets i.i.d data samples $\mathcal{X}, \mathcal{X}_1, ..., \mathcal{X}_N$ as input and outputs the corresponding latent vectors $\mathcal{Z}, \mathcal{Z}_1, ..., \mathcal{Z}_N$. Assume we don't know the distribution of $\mathcal{X}$ but we want to impose an arbitrary distribution $\mathcal{P}$ on $\mathcal{Z}$. This is because we like to think of the decoder as a generative network with prior $\mathcal{P}$. The encoding function $Q_\theta(\mathcal{Z}|\mathcal{X})$ induces aggregated posterior distribution $Q_\theta(\mathcal{Z})$ formulated in eq. 1:

$$Q_\theta(\mathcal{Z}) = \mathbb{E}_\mathcal{X}[Q_\theta(\mathcal{Z}|\mathcal{X})] \tag{1}$$

The decoder and encoder networks of a standard AE can be trained together by minimizing the difference between the original data and its reconstruction. We can impose $\mathcal{P}$ on the latent code vector through regularization of a standard AE by minimizing the discrepancies between aggregated posterior $Q_\theta(\mathcal{Z})$ and an arbitrary prior $\mathcal{P}$. This can be formalized as the following optimization problem:

$$\min_{\theta \in \Theta} \delta(\mathcal{P}, Q_\theta(\mathcal{Z})) \tag{2}$$

where $\delta$ is a suitable discrepancy measure. In other words, parameters of the encoder $\theta$ should be optimized s.t the transported randomness from $\mathcal{X}$ match up to that of prior $\mathcal{P}$. Achievement of a solution for eq. 2 via deterministic functions is guaranteed given enough stochasticity in $\mathcal{X}$, thanks to the Lemma 2.22 in [11]. However, lack of enough stochasticity in $\mathcal{X}$ leads to not smooth enough aggregated posterior $Q_\theta(\mathcal{Z})$ which makes the minimization in Eq. 2 non-trivial. This is specially important considering the deterministic functionality of encoder. It is because the only source of randomness originates from the training data itself, which may not be enough for smoothing out the aggregated posterior.

### 2.1 INTRODUCING RANDOMNESS

Some ways of introducing randomness for smoothing out the aggregated posterior $Q_\theta(\mathcal{Z})$ include:

- Introducing the stochasticity $\mathcal{N}$ at the input or output of the encoder function:

$$Q_\theta(\mathcal{Z}) = \mathbb{E}_\mathcal{X}\mathbb{E}_\mathcal{N}[Q_\theta(\mathcal{Z}, \mathcal{N}|\mathcal{X})] \tag{3}$$

$$Q_\theta(\mathcal{Z}) = \mathbb{E}_\mathcal{X}\mathbb{E}_\mathcal{N}[Q_\theta(\mathcal{Z}|\mathcal{X}, \mathcal{N})] \tag{4}$$

  This may ask for re-parametrization tricks [2].
- Dropouts [9]

Different ways of introducing and exploiting randomness lead to different training dynamics. The choice of discrepancy measure $\delta$ itself leads to various types of training dynamics. Our proposal is to introduce randomness at the heart of the discrepancy measure itself. First, we discuss two known discrepancy measures in the next section.

### 2.2 DISCREPANCY MEASURE

Adversarial autoencoder can be recovered with the following Jenson-Shannon divergence (JSD) choice for the discrepancy measure:

$$\delta_{AN}(\mathcal{P}, Q_\theta(\mathcal{Z})) =$$
$$\max_\alpha \mathbb{E}[\log D_\alpha(\mathcal{Y}) + \log(1 - D_\alpha(\tilde{\mathcal{Y}}))] \tag{5}$$

where $\mathcal{Y} \sim \mathcal{P}$ and $\tilde{\mathcal{Y}} \sim Q_\theta(\mathcal{Z})$ and $D_\alpha$ is the adversary with parameter $\alpha$. Moreover $\mathbb{E}$ is w.r.t to the associated random variable of each term. A straightforward modification is to replace the adversarial network by maximum mean discrepancy (MMD) net [4-5] by assuming the following discrepancy measure:

$$\delta_{MMD}(\mathcal{P}, Q_\theta(\mathcal{Z})) = \sup_{f \in \mathcal{H}} \mathbb{E}[f(\mathcal{Y})] - \mathbb{E}[f(\tilde{\mathcal{Y}})] \tag{6}$$

where $f$ is a function living in a reproducing kernel Hilbert space RKHS $H$ and $\mathcal{Y} \sim \mathcal{P}$ and $\tilde{\mathcal{Y}} \sim Q_\theta(\mathcal{Z})$ as above.

Due to the reproducing property of $H$, the expectation of any function $f$ in RKHS $H$ with respect to random variable $\mathcal{Y}$ can be computed as an inner product with its so called *kernel mean embedding* $\mathbb{E}[k(\mathcal{Y}, .)]$:

$$\mathbb{E}[f(\mathcal{Y})] = \langle f, \mathbb{E}[k(\mathcal{Y}, .)] \rangle \tag{7}$$

$k$ being the kernel associated with RKHS $H$.

**Definition 1** A kernel $k(x, x') : \mathcal{X} \times \mathcal{X} \to \mathbb{R}$ is positive definite (PD) when for all $n > 1$ and $x_1, x_2, .., x_n \in \mathcal{X}$ and $c_1, ..., c_n \in \mathbb{R}$, we have $\sum_{i,j} c_i c_j k(x_i, x_j) \geq 0$.

Gretton [12] demonstrates that a closed form solution exists for Eq. 6 with the following unbiased estimator :

$$MMD_{\text{unbiased}}(H, \tilde{\mathcal{Y}}, \mathcal{Y}) = \frac{1}{N(N-1)} \sum_{n \neq n'} k(\tilde{y}_n, \tilde{y}'_n) +$$

$$\frac{1}{M(M-1)} \sum_{m \neq m'} k(y_m, y'_m) - \frac{2}{MN} \sum_{m=1}^{M} \sum_{n=1}^{N} k(\tilde{y}_n, y_m) \tag{8}$$

We refer to a probabilistic AE that uses Eq. 8 as discrepancy measure MMD-AE.

# 3 DOUBLY STOCHASTIC KERNEL MACHINES AS ADVERSARY

The existence of closed form expression in Eq. 8 makes the implementation of MMD-AE straightforward, however, it hides the *minimax* nature of the problem. If we replace eq. 6 in eq. 2, we are back to a minimax problem similar to that of adversarial networks as the following:

$$\min_\theta \sup_{f \in \mathcal{H}} \boxed{\mathbb{E}_{\mathcal{Y} \sim \mathcal{P}}[f(\mathcal{Y})] - \mathbb{E}_{\tilde{\mathcal{Y}} \sim Q_\theta(\mathcal{Z})}[f(\tilde{\mathcal{Y}})]}_{\delta(\mathcal{P}, Q_\theta(\mathcal{Z}))} \tag{9}$$

Here is where the discussed problem with the lack of enough stochasticity of $\mathcal{X}$ and non-smooth aggregated posterior gets more clear. Using Eq. 7 and Eq. 9, we note that adversary's best response training dynamic is determined by the stochastic gradient terms $\frac{\partial \delta}{\partial f} = \xi(.) := \mathbb{E}[k(\mathcal{Y}, .) - k(\tilde{\mathcal{Y}}, .)]$. Since prior $\mathcal{P}$ is smooth enough, term $\mathbb{E}_{\tilde{\mathcal{Y}} \sim Q_\theta(\mathcal{Z})}[k(\tilde{\mathcal{Y}}, .)]$ determines the smoothness of the stochastic gradients. Not enough stochasticity in $\mathcal{X}$ implies non-smooth gradients and therefore more difficult optimization problem. We propose to massage the stochastic gradients $\xi(.)$ with *extra* source of stochasticity $\mathcal{W}$. This leads to *doubly stochastic* gradient terms $\zeta(.) := \mathbb{E}_{\mathcal{W}}\xi(.)$. Introducing new source of randomness to $\xi(.)$ becomes feasible thanks to the existing duality between the Kernel and Random processes according the following Theorem:

**Theorem 1 [15]** *Duality between Kernels and Random Processes*: If $k(x, x')$ is a PD kernel, then there exits a set $\Omega$, a measure $\mathbb{P}$ on $\Omega$, and random function $\phi_{\mathcal{W}}(x) : \mathcal{X} \to \mathbb{R}$ from $L_2(\Omega, \mathbb{P})$, such that $k(x, x') = \int_\Omega \phi_{\mathcal{W}}(x)\phi_{\mathcal{W}}(x')d\mathbb{P}(\mathcal{W})$.

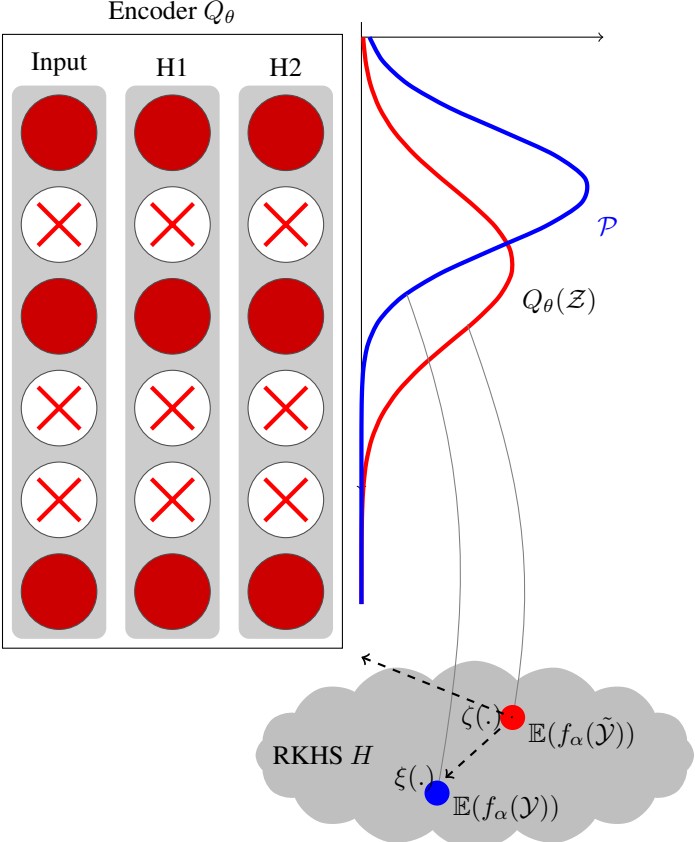

Figure 1: **Scheme of DS-AAE** Each distribution is mapped into a reproducing kernel Hilbert space via an expectation operation. The generator strategy is to adjust the parameter of encoder $\theta$ to decrease the distance between the blue and red dot while adversary's is to adjust $\alpha$ to increase the distance. Not enough stochasticity in $\mathcal{X}$ limits the adversary power as is reflected in its stochastic gradient terms $\xi = \mathbb{E}_{\mathcal{Y} \sim Q_\theta(\mathcal{Z})}[k(\mathcal{Y}, .)]$. With the introduction of new source of randomness, the adversary gains extra power, as is reflected in its doubly stochastic gradient terms $\zeta = \mathbb{E}_\mathcal{W}\xi(.)$. This extra boost to adversary, however, can lead to the search outside of $H$ where the gradients are no longer valid closed form expressions. Fortunately, [6] shows that with small learning rates, the search returns back to $H$ and convergence is guaranteed.

**Example** For Gaussian RBF kernel, $k(x - x') = \exp(\|x - x'\|^2/2\sigma^2)$, $\phi_\mathcal{W}(x) = \exp(-i\mathcal{W}^\top x)$ and $\mathbb{P}(w) = \frac{\exp(-\|w\|_2^2/2)}{(2\pi)^{d/2}}$

As the result of Theorem 1, we can rewrite the massaged gradients as $\zeta = [\phi_\mathcal{W}(\mathcal{Y}) - \phi_\mathcal{W}(\tilde{\mathcal{Y}})]\phi_\mathcal{W}(.)$ with $\mathcal{W} \sim \mathbb{P}(\mathcal{W})$.

### 3.1 APPROXIMATING ADVERSARY $f$ BY ITS DOUBLY STOCHASTIC GRADIENT TERMS

Using a similar approach to Doubly Stochastic Kernel Machines [6], we can now approximate a new adversary with parameter $\alpha$ by the linear combination of doubly stochastic gradient terms $\zeta$ i.e. $f_\alpha(.) = \alpha\zeta(.)$. However, we modify the adversary's objective function to $-[\phi_\mathcal{W}(\tilde{\mathcal{Y}})\phi_\mathcal{W}(.)]_{\tilde{\mathcal{Y}} \sim Q_\theta(\mathcal{Z})}$. This modification does not affect the fixed points of the best response dynamics but provides much stronger gradients early in learning. GAN [3] applies the same modification to the objective function of the adversary. The adversary affects the dynamic of training though the adjustment in $\alpha$. We refer this probabilistic autoencoder as Doubly Stochastic AAE (DS-AAE).

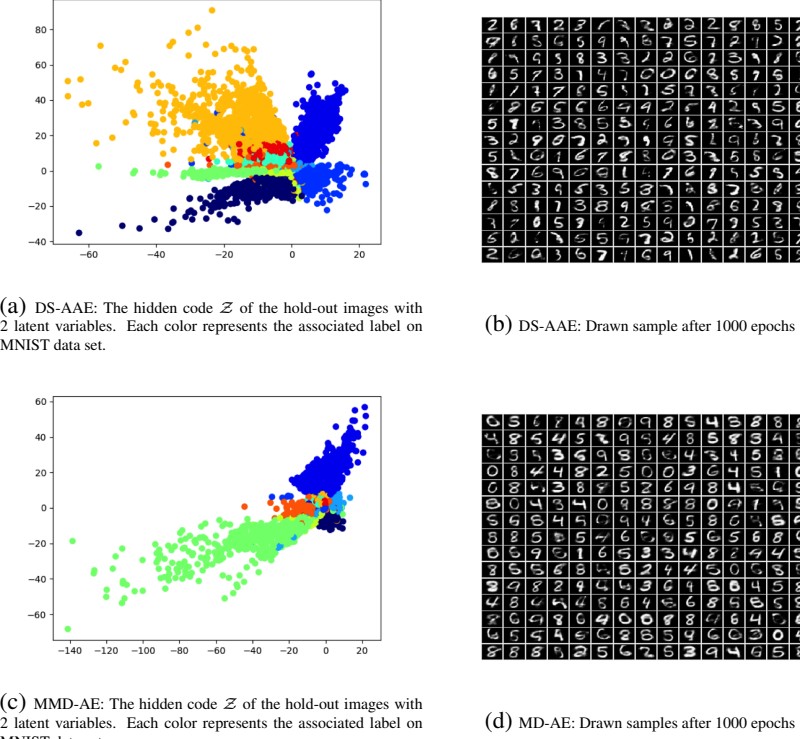

(a) DS-AAE: The hidden code $\mathcal{Z}$ of the hold-out images with 2 latent variables. Each color represents the associated label on MNIST data set.

(b) DS-AAE: Drawn sample after 1000 epochs

(c) MMD-AE: The hidden code $\mathcal{Z}$ of the hold-out images with 2 latent variables. Each color represents the associated label on MNIST data set.

(d) MD-AE: Drawn samples after 1000 epochs

Figure 2: Comparison between MMD-AE and DS-AAE on MNIST.

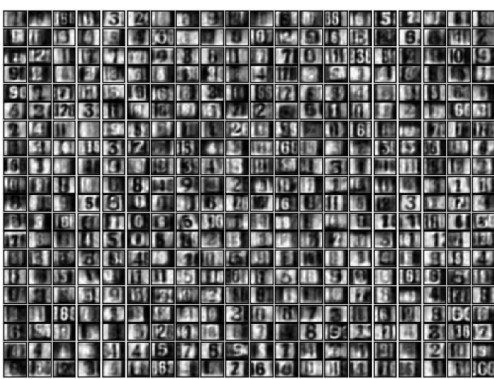

Figure 3: SVHN drawn samples after 1000 epochs

Beside the residual error of approximating $f$ with its first order gradient terms, there is another concern that needs to be addressed. The key difference between $\zeta(.)$ and $\xi(.)$ is that $\zeta(.)$ could fall outside of the RKHS but $\xi(.)$ is always in RKHS. This is due to the term $\phi_w(.)$ not being in RKHS. Fortunately proof of the convergence with small enough learning rate is discussed in [6].

The scheme of DS-AAE is visualized in Fig. 1. To sum up, we proposed replacing the adversary with a space $H'$ of stochastic functions, which may not reside in RKHS $H$ but their convergence to the optimal functions in $H$ is guaranteed. This replacement introduces a new source of randomness,

|  | MNIST |
|---|---|
| GAN [2] | $225 \pm 2$ |
| GMMN + AE [5] | $282 \pm 2$ |
| Adversarial Autoencoder [1] | $340 \pm 2$ |
| MMD-AE | $\mathbf{228 \pm 1.59}$ |
| DS-AAE | $\mathbf{243.16 \pm 1.65}$ |

Table 1: Parzen window estimate of the log-likelihood obtained by drawing 10K samples from the trained model.

which can be considered as a continuous control for encouraging explorations. It prevents the adversary from fitting too closely to the generator and therefore leads to more diverse set of generated samples. Consequently, the decoder serves as a better generative network which unlike MMD-AE scales linearly with the amount of data. We provide empirical evidence on how this replacement outperforms the pre-existing architectures in the next section.

## 4 EXPERIMENTS

We generate $\mathcal{W}$ according to Example 1 with a fixed seed. The encoder and decoder both have 3 layers of 1024, 512, 216 hidden units with ReLU activation for every layer except the last layer of decoder that is a sigmoid activation function. Cross entropy is used for reconstruction loss. The mini-batch size is 1000. The prior $\mathcal{P}$ is Gaussian and the dimensionality of the hidden code is 6 for the DS-AAE and 4 for MMD-AE. The only used dropout is at the first input layer of the encoder with the rate of $20\%$. Initial learning rate for the reconstruction loss is adjusted at 0.001 and 0.001 for the adversarial architectures, followed by Adam stochastic optimization [13].

Comparison of deep generative models is hard especially for the log-likelihood-free models [7]. Parzen window estimation of the log-likelihood is obtained by drawing 10K samples from the trained model on MNIST. The results are shown in Table 1. From the qualitative perspective, we can see from Fig. 2b, Fig. 2d and reported results in [1] that the drawn samples for both MMD-AE and AAE are more homogenous than DS-AAE. In the case of DS-AAE, it is almost as if different persons were writing the digits in each panel. This quality test is also used in [8]. This is because DS-AAE enjoys from extra randomness in the minimax optimization framework, which helps the generative model to explore multiple modes and mitigate the risk of collapse.

The learned coding space of DS-AAE exhibits sharp transitions and has no "holes", as is shown in Fig. 2 (a). This is important to ensure that generating from any part of prior space results in meaningful samples. From this perspective, DS-AAE is similar to AAE and unlike VAE. However it recovers more of a mixture of 2D-Gaussians rather than a 2D-Gaussian distribution. We leave further investigation of this interesting observation to future version of this paper.

SVHN experiment is carried out by the the same architecture and batch size. However, we have used batch-normalization [14] in all the autoencoder layers including the Softmax layer and number of latent codes is 10 in this case. We observe that the generated samples from DS-AA are more diverse, yet they are blurrier than those of MMD-AE, AAE and the rest of GAN architectures. Results are shown in Fig.3.

## 5 CONCLUSION

The recent proposal of generative adversarial models, trades the complexities of sampling algorithms with the complexities of searching Nash equilibrium in minimax games. From this vantage point, role of a machine learning researcher is similar to that of a *mechanism designer*, who intervenes with the dynamic of game through design of suitable loss function. A recent development of probabilistic autoencoder enjoys this framework by placing such minimax games at the output of an encoder to force a desirable prior. We propose a different minimax game and loss function. As a mechanism designer, we intervene with the dynamic of training through introducing suitable randomness to the

loss functions of such minimax games. This makes the game end up in better regions and as a result, better probabilistic autoencoders can be achieved.

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
