# OpenReview forum: "DOUBLY STOCHASTIC ADVERSARIAL AUTOENCODER"
_ICLR.cc/2018/Conference — Reject_

### Official Review · AnonReviewer3 · 2017-11-24
**The paper is not mature enough to be accepted**

**Rating:** 3
**Confidence:** 4

**Review:**

Thank you for the feedback, and I have read it.

The authors claimed that they used techniques in [6] in which I am not an expert for this. However I cannot find the comparison that the authors mentioned in the feedback, so I am not sure if the claim is true.

I still recommend rejection for the paper, and as I said in the first review, the paper is not mature enough.

==== original review ===

The paper describes a generative model that replaces the GAN loss in the adversarial auto-encoder with MMD loss. Although the author claim the novelty as adding noise to the discriminator, it seems to me that at least for the RBF case it just does the following:
1. write down MMD as an integral probability metric (IPM)
2. say the test function, which originally should be in an RKHS, will be approximated using random feature approximations.

Although the authors explained the intuition a bit and showed some empirical results, I still don't see why this method should work better than directly minimising MMD. Also it is not preferred to look at the generated images and claim diversity, instead it's better to have some kind of quantitative metric such as the inception score.

Finally, given the fact that we have too many GAN related papers now, I don't think the innovation contained in the paper (which is using random features) is good enough to be published at ICLR. Also the paper is not clearly written, and I would suggest better not to copy-past paragraphs in the abstract and intro.

That said, I would welcome for the authors feedback and see if I have misunderstood something.

---

> ### Public Comment · (anonymous) · 2017-12-06
> **Thank your comments.**
>
> We would like to thank the reviewer for the kind comments.
>
>     -> Finally, given the fact that we have too many GAN related papers now I don't think the innovation contained in the
>        paper (which is using random features) is good enough to be published at ICLR.
>
>
> Although we use the random feature approximation technique, but it is different from the well known paper of "Random Features for Large-Scale Kernel Machines, A. Rahimi, B, Recht". In fact, we did run experiments for this vanilla case (using random feature approximation) and did NOT lead to promising results. This case would be no different from the MMD case, which avoids the minimax nature of the problem altogether, as is explained in the first paragraph of section 3 of our paper. It comes with the extra benefit of linear computations but with no extra stochasticity.  Our approach is more related to the doubly stochastic kernel machines, ref [6] cited in the paper.
>
>       -> I still don't see why this method should work better than directly minimizing MMD.
>
> The improvement of DS-AAE is because of the extra stochasticity introduced into the architecture.  The adversary’s strategies are stochastic functions which then inject extra stochasticity into the architecture. To back up our assertion on the extra stochasticity empirically, please refer to Fig. 2a. Please note that the proposed approach helps the encoder to recover a mixture of 2D-Gaussians despite having a 2D-Gaussian distribution as the prior.
>
> We do hope our further explanations clear any confusion left in the paper and that the novelty behind DS-AAE design would be more clear.

---

### Official Review · AnonReviewer2 · 2017-11-28
**a straightforward extension of existing algorithms**

**Rating:** 3
**Confidence:** 5

**Review:**


In this paper, the authors propose doubly stochastic adversarial autoencoder, which is essentially applying the doubly stochastic gradient for the variational form of maximum mean discrepancy.

The most severe issue is lacking novelty. It is a straightforward combination of existing work, therefore, the contribution of this work is rare.

Moreover, some of the claims in the paper are not appropriate. For example, using random features to approximate the kernel function does not bring extra stochasticity. The random features are fixed once sampled from the base measure of the corresponding kernel. Basically, you can view the random feature approximation as a linear combination of fixed nonlinear basis which are sampled from some distribution.

Finally, the experiments are promising. However, to be more convincing, more benchmarks, e.g., cifar10/100 and CelebA, are needed.

---

> ### Public Comment · (anonymous) · 2017-12-06
> **Thank you for your comments.**
>
> We hope our further explanations clear any confusion left in the paper.
>
>  > Moreover, some of the claims in the paper are not appropriate. For example, using random features to approximate
>     the kernel function does not bring extra stochasticity. The random features are fixed once sampled from the base
>    measure of the corresponding kernel. Basically, you can view the random feature approximation as a linear
>   combination of fixed nonlinear basis which are sampled from some distribution.
>
> Although we use the random feature approximation technique, but it is different from the well known paper of "Random Features for Large-Scale Kernel Machines, A. Rahimi, B, Recht". In fact, we did run experiments for this vanilla case (using random feature approximation) and did NOT lead to promising results. This case would be no different from the MMD case, which avoids the minimax nature of the problem altogether, as is explained in the first paragraph of section 3 of our paper. It comes with the extra benefit of linear computations but with no extra stochasticity. We totally agree.
>
> Our approach is more related to the doubly stochastic kernel machines, ref [6] cited in the paper.  The introduced stochasticity is then the result of the stochastic functions as the adversary’s strategies. To back up our assertion on the extra stochasticity empirically, please refer to Fig. 2a. Please note that the proposed approach helps the encoder to recover a mixture of 2D-Gaussians despite having a 2D-Gaussian distribution as the prior.
>
> We do hope the novelty of approach would be more clear after these comments.

---

### Official Review · AnonReviewer1 · 2017-11-28
**Clear reject**

**Rating:** 2
**Confidence:** 5

**Review:**

This manuscript explores the idea of adding noise to the adversary's play in GAN dynamics over an RKHS. This is equivalent to adding noise to the gradient update, using the duality of reproducing kernels. Unfortunately, the evaluation here is wholly unsatisfactory to justify the manuscript's claims. No concrete practical algorithm specification is given (only a couple of ideas to inject noise listed), only a qualitative one on a 2-dimensional latent space in MNIST, and an inconclusive one using the much-doubted Parzen window KDE method. The idea as stated in the abstract and introduction may well be worth pursuing, but not on the evidence provided by the rest of the manuscript.

---

> ### Public Comment · (anonymous) · 2017-12-06
> **Thank you for your comments.**
>
> Thank you for the comments. We hope our further explanations clear any confusion left in the paper.
>
>      -> This manuscript explores the idea of adding noise to the adversary's play in GAN dynamics over an RKHS. This is
>      equivalent to adding noise to the gradient update, using the duality of reproducing kernels.
>
> The approach is not equivalent to adding noise to the gradient update. The introduced stochasticity is the result of the stochastic functions as the adversary’s strategies. The introduced approach, however, can be perceived as a mechanism for smoothing the gradients. This is to mitigate the model collapse issue. In order to see how DS-AAE can address the mode collapse issue, please consider a case when there is a "hole" in the learned coding space (which would be expected in the course of training - The learned coding space is also visualized in Fig.2a and Fig. 2c after training).  In such cases, the adversary cannot discriminate against the boundaries around the "hole" properly.  This is because of the bumpy gradients terms. This leads to mode collapse issue, discussed at the introduction and is indeed the main motivation for proposing DS-AAE. The bumpy gradient terms can be avoided using DS-AAE architecture. This is mathematically explained at the bottom of page 3, right before Theorem 1.
>
>     -> No concrete practical algorithm specification is given (only a couple of ideas to inject noise listed), only a qualitative
>     one on a 2-dimensional latent space in MNIST, and an inconclusive one using the much-doubted Parzen window KDE
>     method. T
>
> Dimensionality of the hidden codes are 6 and 4 for the Fig.2b  and Fig. 2d, respectively. Only figures 2.a and 2.c are on 2-dimensional latent space (for visualization purposes). More importantly, Fig. 2a shows that the introduced stochastically helps the encoder to recover a mixture of 2D-Gaussians despite having a 2D-Gaussian distribution as prior.

---

### Decision · Program_Chairs · 2018-01-29
**ICLR 2018 Conference Acceptance Decision**

**Decision:**

Reject

**Comment:**

The reviewers all outlined concerns regarding novelty and the maturity of this work. It would be helpful to clarify the relation to doubly stochastic kernel machines as opposed to random kitchen sinks, and to provide more insight into how this stochasticity helps. Finally, the approach should be tried on more difficult image datasets.